# Groundwater quality characterization using an integrated water quality index and multivariate statistical techniques

Vinay Kumar Gautam[1,2]*, Mahesh Kothari[1], Baqer Al-Ramadan[3,4], Pradeep Kumar Singh[1], Harsh Upadhyay[1,2], Chaitanya B. Pande [2,5]*, Fahad Alshehri[2], Zaher Mundher Yaseen[6,7]

**1** Department of Soil and Water Engineering, Maharana Pratap University of Agriculture & Technology, Udaipur (Rajasthan), India, **2** Abdullah Alrushaid Chair for Earth Science Remote Sensing Research, Geology and Geophysics Department, College of Science, King Saud University, Riyadh, Saudi Arabia, **3** Architecture & City Design Department, King Fahd University of Petroleum & Minerals, Dhahran, Saudi Arabia, **4** Interdisciplinary Research Center for Smart Mobility and Logistics, King Fahd University of Petroleum & Minerals, Dhahran, Saudi Arabia, **5** New Era and Development in Civil Engineering Research Group, Scientific Research Center, Al-Ayen University, Thi-Qar, Nasiriyah, Iraq, **6** Civil and Environmental Engineering Department, King Fahd University of Petroleum & Minerals, Dhahran, Saudi Arabia, **7** Interdisciplinary Research Center for Membranes and Water Security, King Fahd University of Petroleum & Minerals, Dhahran, Saudi Arabia

* chaitanay45@gmail.com (CBP); drvgautam95@gmail.com (VKG)

**Data Availability Statement:** All relevant data are within the paper.

**Funding:** This study was financially supported by Abdullah Alrushaid Chair for Earth Science Remote

## Abstract

This study attempts to characterize and interpret the groundwater quality (GWQ) using a GIS environment and multivariate statistical approach (MSA) for the Jakham River Basin (JRB) in Southern Rajasthan. In this paper, analysis of various statistical indicators such as the Water Quality Index (WQI) and multivariate statistical methods, i.e., principal component analysis and correspondence analysis (PCA and CA), were implemented on the pre and post-monsoon water quality datasets. All these methods help identify the most critical factor in controlling GWQ for potable water. In pre-monsoon (PRM) and post-monsoon (POM) seasons, the computed value of WQI has ranged between 28.28 to 116.74 and from 29.49 to 111.98, respectively. As per the GIS-based WQI findings, 63.42 percent of the groundwater samples during the PRM season and 42.02 percent during the POM were classed as 'good' and could be consumed for drinking. The Principal component analysis (PCA) is a suitable tool for simplification of the evaluation process in water quality analysis. The PCA correlation matrix defines the relation among the water quality parameters, which helps to detect the natural or anthropogenic influence on sub-surface water. The finding of PCA's factor analysis shows the impact of geological and human intervention, as increased levels of EC, TDS, $Na^+$, $Cl^-$, $HCO_3^-$, $F^-$, and $SO_4^{2-}$ on potable water. In this study, hierarchical cluster analysis (HCA) was used to categories the WQ parameters for PRM and POR seasons using the Ward technique. The research outcomes of this study can be used as baseline data for GWQ development activities and protect human health from water-borne diseases in the southern region of Rajasthan.

Sensing Research at King Saud University, Riyadh, Saudi Arabia. The funder provided supported for field data collection and analysis. No additional external funding was received for this study. The funder had no additional role in study design, data collection and analysis, decision to publish, or preparation of the manuscript.

**Competing interests:** The authors have declared that no competing interests exist.

## 1. Introduction

Adequate water supply is one of the most common concern nowadays for every society [1,2]. This leads to sustainable management of water resources at the stakeholder end, and a dedicated approach is required to achieve this cutting-edge concept [3]. Under the umbrella of environmental science, high-accuracy monitoring systems with low error have been developed [4,5]. It is a significant concern, especially for developing countries facing water quality issues for agriculture and daily human living [6,7]. In India, a chunk of the population is dependent upon rivers and lakes for the fulfillment of their water requirement, though, on the other hand, sewage and discharged water management is very poor [8]. As a result, the available water is prone to contamination and affecting human health; it is necessary to monitor available water resources [9]. The GWQ is affected by a wide variety of natural and anthropogenic factors [10]. The rock-water interaction, mineral concentration, and ion exchange are some natural factors [11–13]. According to a UN report, 22% of deaths are directly connected to the waterborne [14,15]. Environmental authorities and the general public can be informed through ecological indicators and pollution indices to raise awareness about safe drinking water [16–18]. As per CGWB [19], two blocks of the JRB basin, namely Pratapgarh and Chhoti Sadri, have been characterized as a sub-critical stage due to high fluctuations in water level and worrying WQ issues before and after the monsoon period [20,21]. Gautam et al. predicted the GWQ for irrigation purposes for Southern Rajasthan using ANN modeling [22]. A total of six input parameters (Na, K, Ca, Mg, CO3, and HCO3) have been identified for analysis. The model predicted the ANN structures ANN4 (3-12-1), ANN4 (4-15-1), ANN1 (4-5-1), and ANN4 (3-12-1), which were found best suited for SAR, %Na, RSC, and KR water quality indicators for the basin. This ANN model of water quality could be helpful in sustainable groundwater management and crop suitability planning. GWQ characterization refers to the physico-chemical and statistical analysis of WQ parameters based on different standardized approaches [23,24]. The GWQ is a significant aspect of the sustainable growth of water resources, as many fields rely on it [25–29]. Gautam et al. assessed the GWQ in the Pratapgarh district of southern Rajasthan and found that only 63% and 42% of the area is suitable for drinking during PRM and POM season [30]. The assessment of GWQ is a hectic task that involves handling many variables, each of which has the potential to exert a specific impact on overall WQ [31]. In Southern Rajasthan, GWQ has not been characterized using this integrated methodology [21,32]. The GWQ indexing is a new approach for determining water potability for human consumption with an index varying on a 0–100 scale, representing the suitability of WQ in a particular space and period. The spatial maps and GWQ can also be generated using GIS and IDW [33–35]. As a response, the WQI has been applied as an essential approach for evaluating the GWQ [36–39].

Correlation The correlation matrix is an essential statistical tool for establishing relationships among the various WQ parameters under the influence of other geological and chemical factors [40,41]. Doza et al. monitored and analyzed the GWQ in the Faridpur district of central Bangladesh of 60 open wells [6]. The correlation matrix (CM) outcomes were consistent with other statistical tool results [42–45]. However, PCA has emerged as a crucial process for application to multivariate datasets concerning WQ parameters [46,47]. As a part of PCA, grouped variables termed principal components (PCs) may be transformed using a statistical method known as factor analysis (FA) [48]. Hence, in this study, the crucial WQ parameters of the local aquifers were assessed using PCA. Gulgundi and Shetty determined the essential groundwater hydro-geochemical patterns, significantly the variance in WQ, found in Bengaluru city, Karnataka [49].

Cluster Analysis (CA) identifies variable groupings by assessing proximity among WQ components based on specific features [50–52]. Combining the WQI index with GIS offers a comprehensive depiction of fluctuations in GWQ traits. Multivariate statistical methods aid in evaluating GWQ, shedding light on natural and anthropogenic influences [53,54]. Utilizing the Ward's approach and squared Euclidean distance, this study generated a dendrogram. Besides multivariate techniques, graphical tools like Box-Whisker plots illustrate mean, maximum, and minimum values in a single view. This research anticipates contributing to GWQ assurance and sustainable water resource management in the JRB basin [55–57]. The study characterizes physio-chemical traits of groundwater from JRB wells, establishing robust statistical techniques for GWQ assessment within a GIS framework.

## 2. Materials and methods

### 2.1 Study area

Geographically, the Jakham River Basin (JRB) is located between the latitudes of 24.451˚N—23.988˚N and the longitudes of 74.501˚E- 74.802˚E, covering an area of 953 km$^2$ in the upper reaches of the Mahi River Basin (Fig 1). The mouth of the Jakham River is located in the hills of the S-W Pratapgarh, Rajasthan. The S-W portion of the basin has a high index of forest and hilly areas; also, the NE-SE part is covered with urban and agricultural practices, which reflects the land use diversity under the basin [58]. The basin also observes a moderate evaporation rate, i.e., approximately 11.20 mm/day in summer. The central part of the basin is involved in the cultivation of opium and processing its by-products, leading to the overuse of fertilizers and saline chemicals. Hence, the north to central part of the basin has a salinity problem in groundwater. Most of the settlement is in a basaltic area extending south of the basin. According to studies, these granite formations are not considered safe water sources. Considerable groundwater potential exists within the contact zone of basalt and other lithological units.

The GW samples were collected from seventy-six well locations during the PRM and POM seasons (2019–2020) (Fig 2). During 2019 and 2020, in pre and post-monsoon seasons (PRM & POM), 76 samples were collected from the open wells located within JRB (Fig 2). FOR STATISTICAL ANALYSIS, other GW quality data of the last 13 years were collected from the Rajasthan Groundwater Department (RGWD) and CGWB, Jaipur. High-quality sealed polyethylene bottles (250 ml) were used for sample collection. Some WQ parameters were analyzed on the spot through a low-cost Rapid Water Quality Testing Kit (RaQKT) kit and another lab [59]. We have used the Charge Balance Error (CBE) method to ensure the precision of our analyzed samples:

$$\text{CBE ratio} = \frac{(\Sigma \text{ Cations} - \Sigma \text{ Anions})}{(\Sigma \text{ Cations} + \Sigma \text{ Anions})} \times 100\% \tag{1}$$

Most of the investigated samples had concentrations of CBE below 10%. The flowchart of the methodology that illustrates the processes involved in the characterization of GWQ is presented in the methodology. This flow chart provides the step by step process for spatial mapping of WQI index and statistical analysis of groundwater parameters through PCA, CA and correlation matrix.

### 2.2 Groundwater Quality Index (GWQI)

The WQI is a common approach to express significant quantities of WQ data as a single numerical value. It indicates the parameters whose index values express the complete water

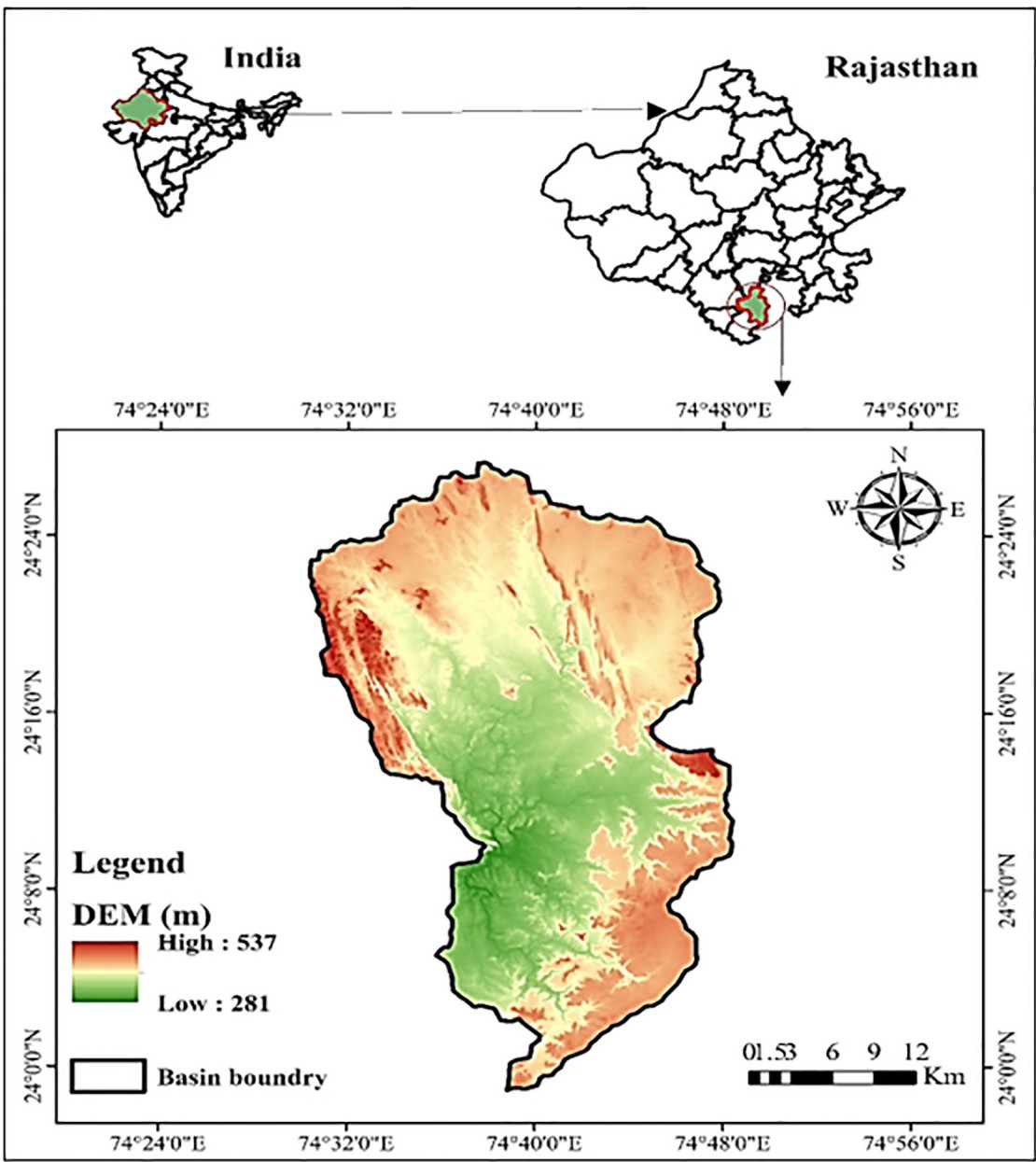

**Fig 1. Map of the JRB Basin with elevation gradient.**

quality. This indexing is suited for the analysis of the DWQ as well as irrigation water. The criteria for WQI mapping are presented in Table 1. Brown et al. [60] formulated the weighted arithmetic indexing (WAI) method, which is often used for estimating WQI. The given Eq 2 has been applied for calculation of WQI is:

$$\mathrm{WQI} = \frac{\left(\sum_{i=1}^{n} q_i w_i\right)}{\left(\sum_{i=1}^{n} w_i\right)} \qquad (2)$$

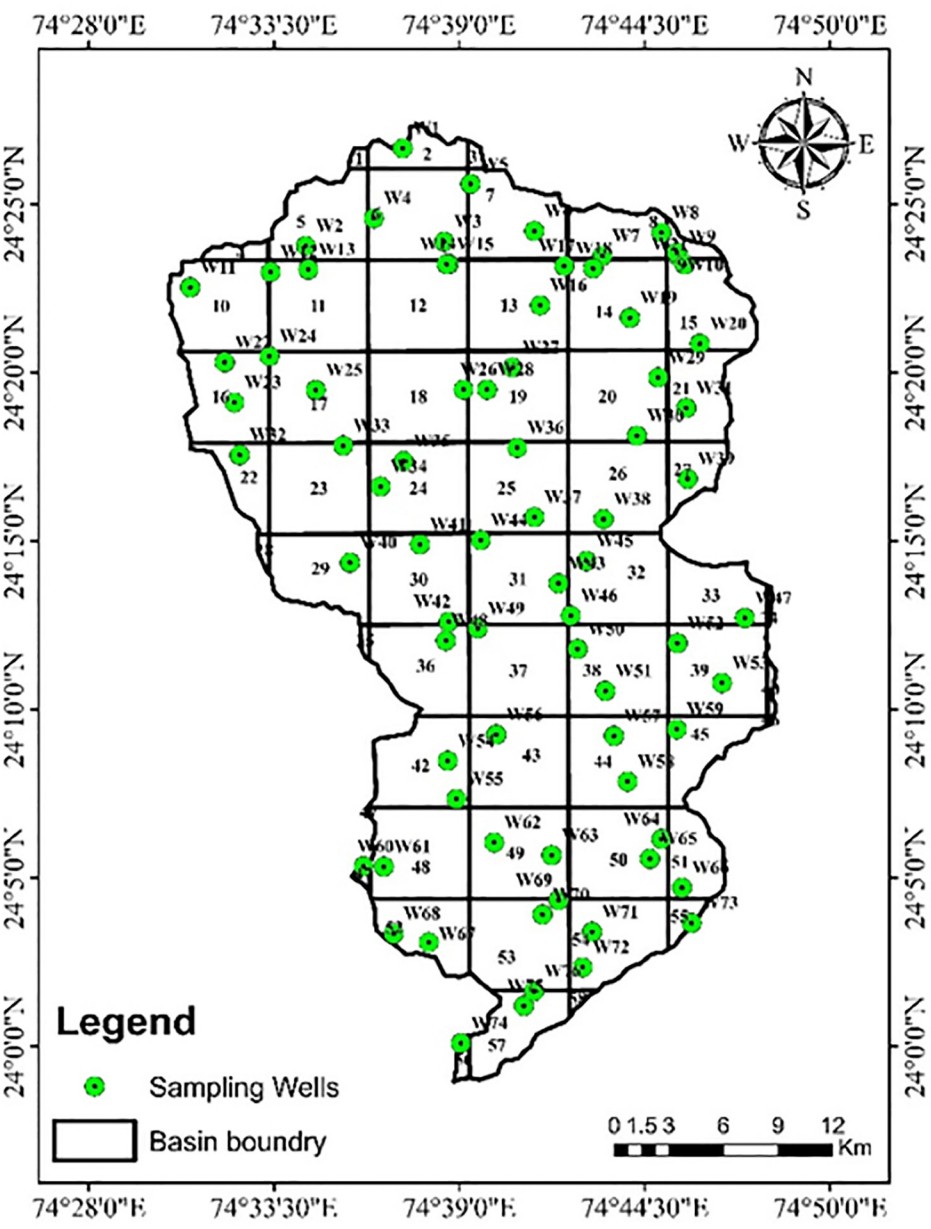

**Fig 2. Sample points of GW wells.**

**Table 1. Criteria for WQI mapping.**

| S.No. | WQI index class | Status |
|---|---|---|
| 1. | I (0–25) | Excellent |
| 2. | II (26–50) | Good |
| 3. | III (51–75) | Poor |
| 4. | IV (76–100) | Very Poor |
| 5. | V (> 100) | Unfit for consumption |

where, $q_i$ = quality rating (sub index) of ith WQ parameter, $w_i$ = unit weight of ith WQ parameter.

The number of reflecting the relative value of the parameter in polluted water in accordance with the standard allowable value is the water quality and quantity ratings ($q_i$), corresponding to the $i^{th}$ parameter.

$$q_i = 100\left(\frac{V_i - V_{io}}{S_i - V_{io}}\right) \tag{3}$$

where, $v_i$ = estimated value of the $i^{th}$ parameter, $v_{io}$ = ideal value of the $i^{th}$ parameter, $s_i$ = standard permissible value of the ith parameter. In most of the cases $v_i = 0$ except in pH.

Unit weight ($W_i$) calculation is to various water quality parameters are inversely proportional to the recommended standards for the corresponding parameters.

$$W_i = \frac{k}{S_i} \tag{4}$$

where, $W_i$ = unit weight for the $i^{th}$ parameter, $S_i$ = standard permissible value for $i^{th}$ parameter, k = proportionality constant.

## 2.3 Implementing multivariate statistical technique for interpretation of GWQ parameters

Different multivariate and statistical methods, such as the box plot, Principal Component Analysis (PCA), and Cluster Analysis (CA), were used to figure out the chemical properties of groundwater. The relationship among quality indicators of groundwater samples can be represented graphically with these tools. These evaluations were performed using the SPSS 26.0 and XLSTAT software.

## 2.4 Box and whisker plot

Box and whisker plots are a prominent statistical tool for visualization and identification of water quality parameters that may significantly influence the water chemistry of a given area. Box and whisker plots depict a particular data distribution's median, range, and shape. A standard box and whisker plot consists of a rectangular box bisected horizontally and vertically with the longer bisector extending outside the box for equal lengths, as shown in (Fig 3) [61]. The shorter bisector in the box represents the median, while the extremes of the extended bisector, also called the whisker, indicate the minimum and maximum of the distribution. The shorter sides of a box and whisker plot represent the 25th and 75th percentile of the statistics computed from the observed data. The length of the box indicates the range of the central 50% of data, whereas the length of the whiskers showcases the range of bottom and top 25% of data. The whisker extends to data extremities within 1.5 times the interquartile range of the data, providing an overview of how the distribution's tails are stretched. It may be noted that the box width holds no meaning; hence, the plot can be narrow without affecting its visual impact.

## 2.5 Principal Component Analysis (PCA)

Principal Component Analysis (PCA) is the most commonly used statistical method employed to interpret GWQ parameters. Implementation of PCA as a multivariate statistical tool is usually carried out to compress water quality datasets spanning multiple dimensions in order to reduce jitter and redundancy and to promote an efficient analysis. PCA approaches datasets of several correlated components by portraying them as smaller sets of independent, uncorrelated

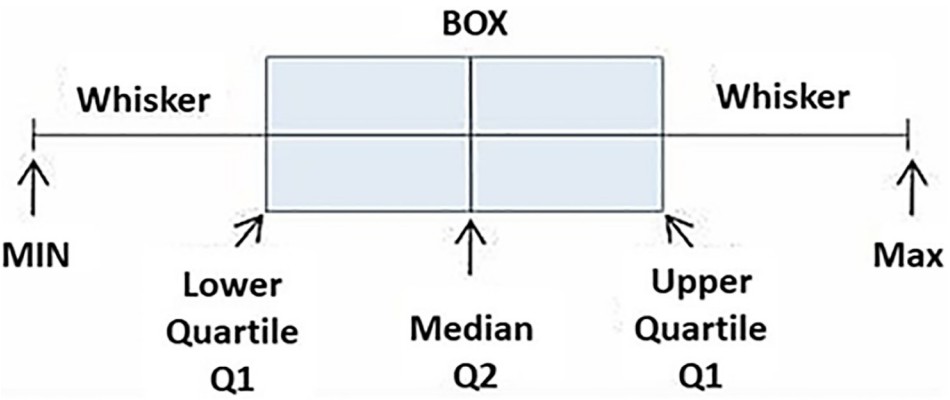

**Fig 3. Representations of box and whisker plot.**

variables. It incorporates data in a correlation matrix and reorganizes them in a system that can improve the interpretability of the authentic structure of the data. The process of PCA commences with generating a novel set of GWQ variables (called PCs) based on linear combinations of variables belonging to the original datasets. Generally, the entire PCA process can be divided into two steps: standardization of data and extracting PCs. Initially, measured water quality data ($X_{ji}$) was standardized by Z-scale transformation using the following formula 5:

$$z = \frac{X_{ji} - \bar{X}_j}{S_j} \tag{5}$$

where, $X_{ji}$ is a value of $j^{th}$ GWQ parameters measured at ith location, $X_j$ is the mean value of $j^{th}$ parameter, $S_j$ is the standard deviation of the jth parameter.

## 2.6 Correlation matrix

The variance proportion of one parameter explained by its relationship with another parameter may be measured using a correlation coefficient. Correlation coefficient varies between -1 and +1, demonstrating extreme perfect dissimilarity and similarity, respectively, while a correlation coefficient equating 0 denotes a lack of any relationship between the variables.

## 2.7 Cluster Analysis (CA)

The cluster analysis technique segregates the observed dataset into various clusters or groups based on their similarities measured concerning their respective correlation coefficients. This multivariate method provides a means for classifying a given data into clusters based on similarity or closeness measures. Cluster analysis is a popular tool usually employed to assess the possibility of grouping GWQ variables for different samples as per their similarities in hydro-chemical characteristics [62]. Each group created represents a particular hydro-chemical process that may occur in the study area. The water quality variables commonly incorporated into CA comprise percent equivalents of major ions, pH, and/or salinity levels [63–66].

## 2.8 Hierarchical Cluster Analysis (HCA)

Hierarchical Cluster Analysis (HCA) was implemented in the present study in the form of a data categorization tool to categorise homogeneous chemical parameters. Relevant literature outlines a multitude of clustering methods among which HCA is most commonly used in

environmental sciences [63]. The theme of this technique is usually associated with identical observations in the datasets. The main purpose of this technique is to group a cluster of more than one parameter. The Euclidean distance approach was applied to find the similarities and dissimilarities in selected variables, i and j, which were computed as follows formula 6:

$$d_{ij}^2 = \sum\nolimits_{k=1}^{m} \left( Z_{(i,k)} - Z_{(j,k)} \right) \tag{6}$$

where, $d_{ij}$ is the Euclidean distance, $Z_{ik}$ and $Z_{jk}$ are the variable, k for objects i and j, respectively, and m is the number of variables [67]. A short distance indicates the closeness of two chemical parameters, while a high distance shows the dissimilarity in the parameters [68–72]. The detailed methdology is presented in Fig 4.

## 3. Result and discussions

### 3.1 GIS based water quality indexing

The WQI map of PRM & POM season was generated with Q-GIS software based on pre-selected GWQ factors to determine the different quality categories, i.e., excellent (Class I), good (Class II), poor (Class III), very poor (Class IV) and unfit for consumption (Class V) at every single well (Fig 5a). The WQI for Jakham River Basin (JRB) in PRM season with 76 samples has been analyzed and categorized, per BIS/WHO standards. The computed WQIs ranged from 30 to 105 within the basin during the PRM season. The analysis showcased that none of the sampling locations belonged to the Class I category. GWQ maps created using Q-GIS software highlighted that during the PRM season, 603.705 km$^2$, i.e., 63.42% of the total basin area, possessed good water quality, followed by 326.02 km$^2$ (34.21% of the total basin area) falling under the poor category of water quality. Furthermore, it was discovered that the basin also consisted of minor percentages of area with WQI under very poor (2.21% of total basin area) and unfit for consumption (0.161% of total basin area) categories. The TDS in water was identified as the most active GWQ parameter, followed by pH, EC, and Na in the PRM season. The remote sensing data can be also helpful for monitoring of water quality [73–75]. The water quality important can be monitored based on google earth engine and satellite data [76–78]. The rainfall and rocks also effect the groundwater quality parameters [79–81].

The GWQ, during POM season, is presented in Fig 5b. As observed in the PRM season, no sampling locations were found in the 'excellent' category. Around 490.89 km$^2$ (51.51%) area of the basin was observed with poor water quality for potable uses, followed by 400.54 km$^2$ (42.02%) area with good water quality. The basin has a percentage of groundwater in the very poor (6.30%) and unfit for consumption (0.16%) category. The central region of the basin appears to have the highest concentration of minerals with desirable quality. The final WQI indicates more area (63.42%) under good WQ rating in the PRM season. Further, it decreased to 42.02% in the post-monsoon season due to geogenic and climatic effects on it. The Na and Ca content were active parameters in the drinking water in the POM season.

### 3.2 Box and whisker plots for PRM and POM season

The Box and Whisker plots of different WQ parameters for the PRM season indicated that WQ parameters exhibit a temporal variation, as depicted in Fig 6a–6t. In Fig 6a to 6t, the (+) sign indicates the mean value, and (*) indicates the maximum/minimum value of the water quality parameters. The XLSTAT-2020 software has been used for preparing these plots. According to Box-whisker plots, parameters EC, Na, K, and Cl didn't show major temporal variations from 2006 to 2020. The GW parameters, such as the ions' type, concentrations, and water temperature, influence the EC values [69,82,83]. In the past years (2006–2009), the pH

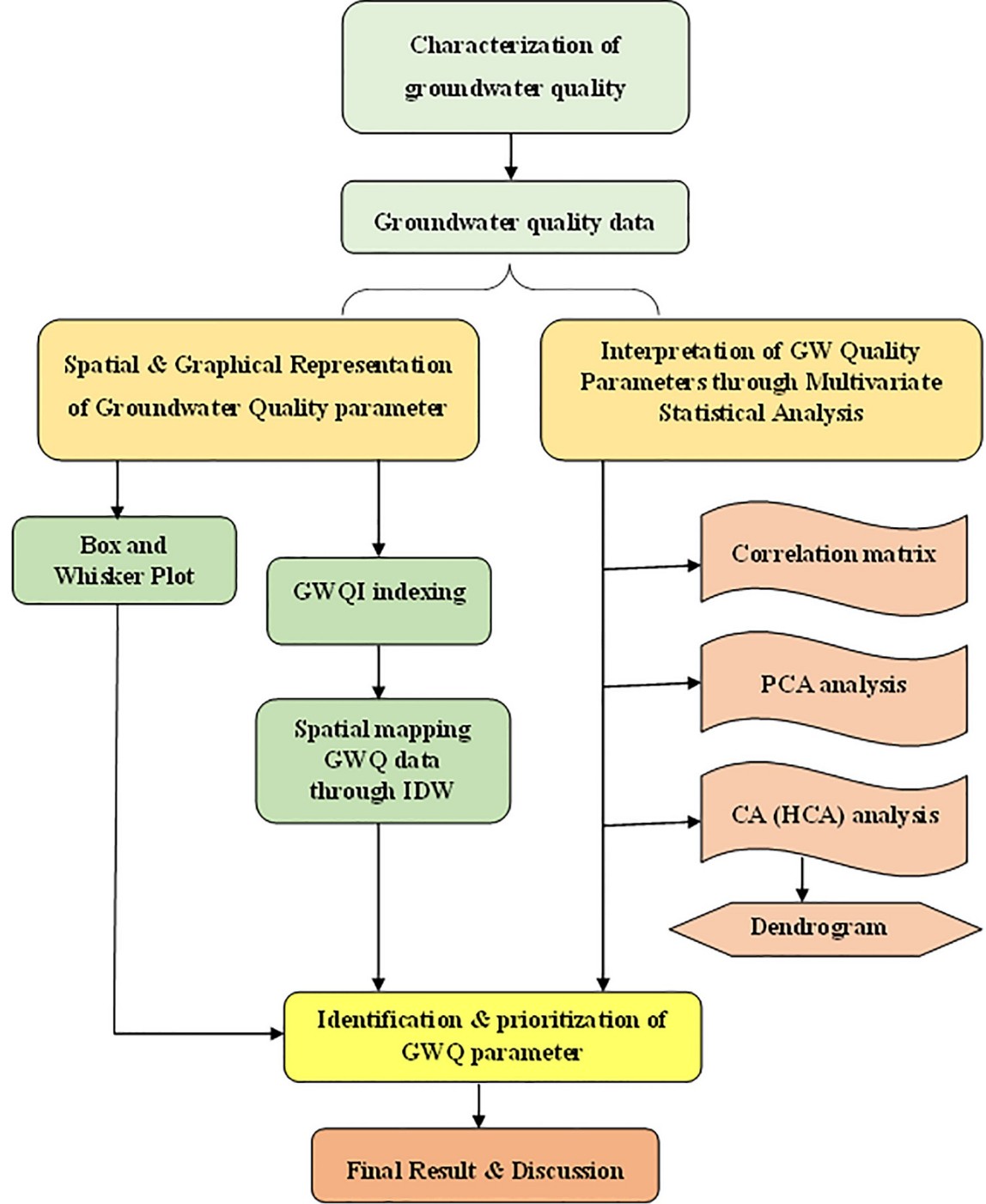

**Fig 4. Flowchart of characterization of GWQ in the study area.**

value had a wider range of data, but it has declined recently. The inter-quartile range (IQR) for pH, TDS, Ca, and HCO3 is high in the box plot graph, indicating that these values fluctuate during the pre-monsoon season. The variation in the TDS mean value shows the geogenic activities (dissolution of rock salt) in the study area [84–86]. The geostatics and rock chemical

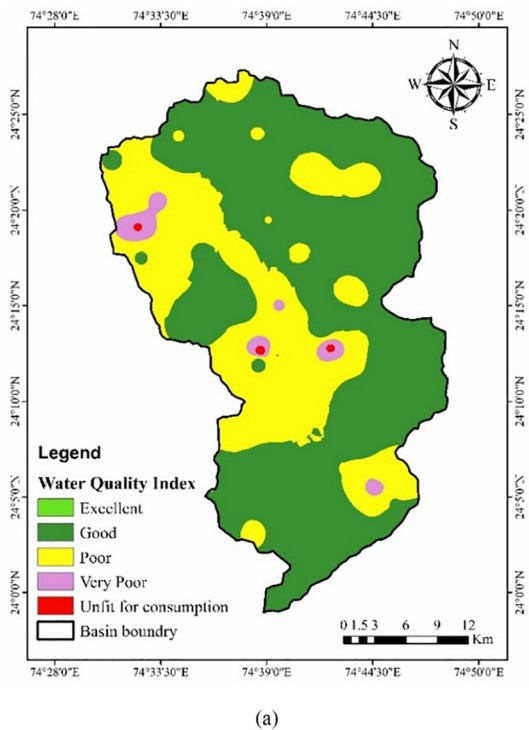

(a)

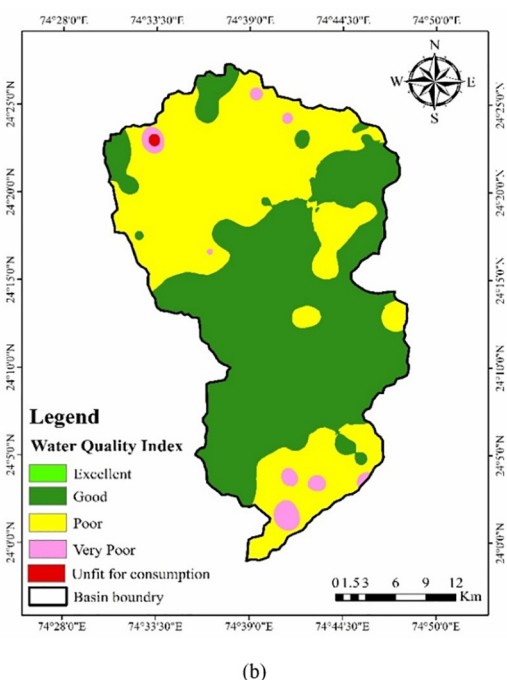

(b)

**Fig 5. GWQ index Map for (a) PRM (b) POM season.**

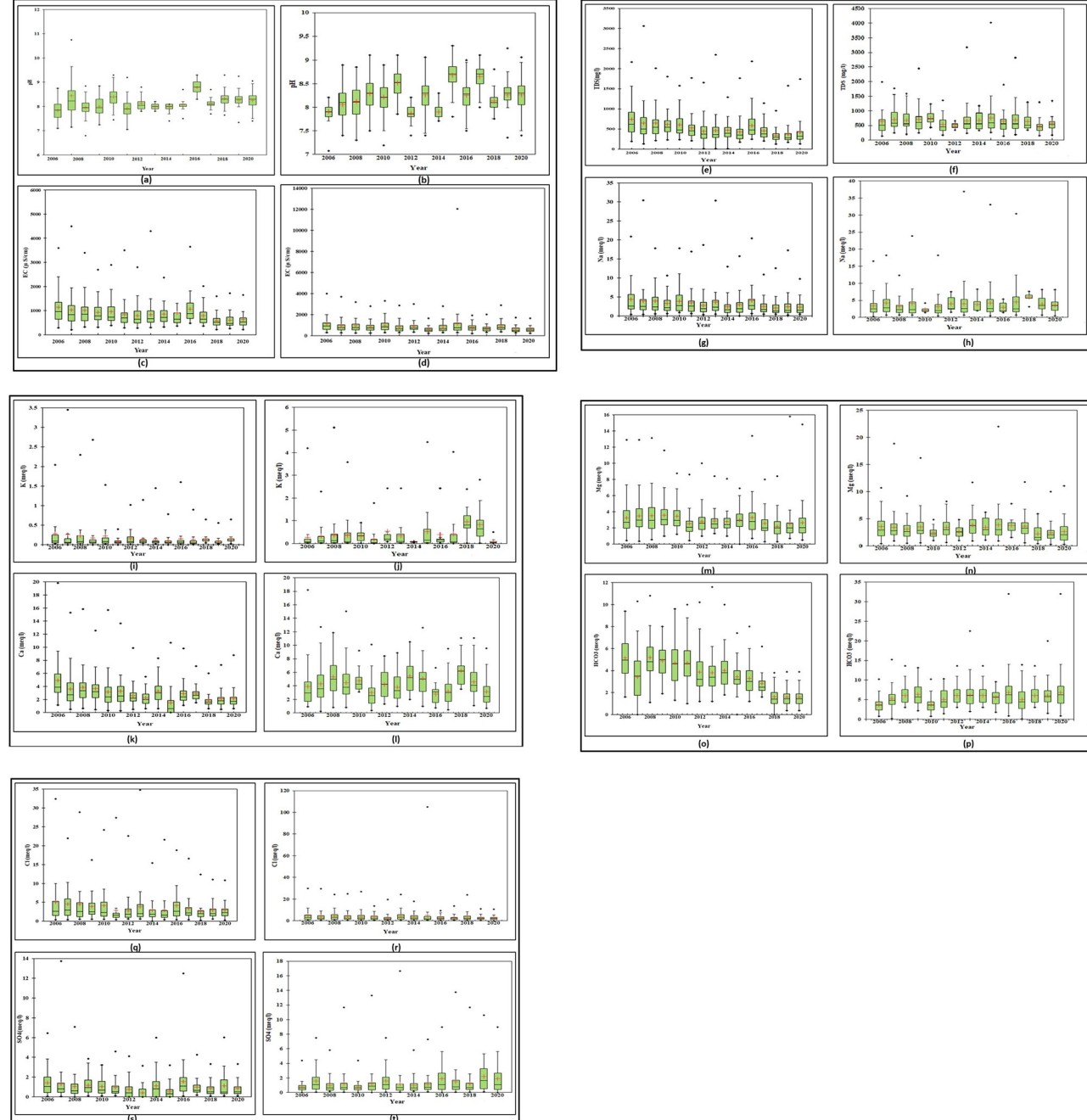

**Fig 6. Box and whisker plots for pre and post monsoon season, a) pH, b) pH, c) EC, d) EC, e) TDS, f) TDS, g) Na, h)Na, i)K, j)K, k) Ca, l)Ca, m) Mg, n)Mg, o)HCO3, p)HCO3, q) Cl, r)Cl, s)SO4, t)SO4 during pre-post monsoon, respectively.**

weathering also more impact on the groundwater quality factors with geostatics tools can more systimatic present the data of water quality [87–89].

Similarly, the Box-whisker plots of post-monsoon parameters EC, Na, Cl, CO3, and HCO3 didn't show the major temporal variations. In the years (2006–2020), the pH value had a wider range of data and more variation in the mean value. The low variation in the Cl mean value

shows less leaching and geogenic activities during POM in the study area [70,90,91]. The box and whisker plots also show some outliers in the data set during the post-monsoon season (2006–2020).

## 3.2 Multivariate statistical analysisa

**3.2.1 Inter-correlation among WQ Parameters for PRM and POM season.** A correction matrix of thirteen GWQ parameters was generated to obtain the inter-correlationship among selected which were used as independent variables, in modelling and characterize the groundwater quality [92–94]. The TDS was observed strong correlation (correlation coefficient greater than 0.9) with Cl and EC, as mentioned in Table 2. Even, other parameters, such as TDS with Na, K, Ca, Mg, $HCO_3$, Cl with Na, K, Ca, Mg, and Na with EC and Mg, have a good correlation (correlation coefficient greater than 0.75). The pH was observed, poorly correlated with Cl and $SO_4$. It was not easy to organize the parameters into components and assign some physical significance at this phase, since certain parameters, such as F, $SO_4$, and $NO_3$, have no significant correlation with any other parameter. The correlation matrix was subjected to the principal component analysis.

In post-monsoon season (POM), TDS was observed good correlation with Na, Ca, and Mg, as mentioned in Table 3. Also, EC has good correlation with Cl. It was not easy to organize the parameters into components and assign some physical significance at this phase, since certain parameters, such as pH, F, $SO_4$, and $NO_3$, $HCO_3$ and $CO_3$ have no significant correlation with any other parameter.

**3.2.2 PCA factor analysis.** The PCA data was verified for analysis with the help of K-M-O and Barlett's test (Table 4). MATLAB 2020 software was used for the above analysis. In PRM and POM season test value was found 0.702 and 0.698 respectively, which indicates data adequacy for factor analyses.

The PCA was applied to a WQ variable's correlation matrix consisting of 13 physio-chemical parameters. Its purpose is to determine the individual loadings of each of the 13 variables that affect WQ characteristics. Eigenvalues (ig) are frequently employed to derive components

**Table 2. Inter-correlations matrix of different water quality parameters for PRM season.**

| Parameter | pH | EC | TDS | Na | K | Ca | Mg | Cl | SO$_4$ | CO$_3$ | HCO$_3$ | NO$_3$ | F |
|---|---|---|---|---|---|---|---|---|---|---|---|---|---|
| **pH** | 1 | | | | | | | | | | | | |
| **EC** | -.011 | 1 | | | | | | | | | | | |
| **TDS** | -.090 | **.913**\*\* | 1 | | | | | | | | | | |
| **Na** | .020 | **.895**\*\* | **.885**\*\* | 1 | | | | | | | | | |
| **K** | .032 | .678\*\* | **.811**\*\* | .660\*\* | 1 | | | | | | | | |
| **Ca** | -.299 | **.724**\*\* | **.898**\*\* | .668\*\* | .743\*\* | 1 | | | | | | | |
| **Mg** | .083 | **.848**\*\* | **.851**\*\* | **.777**\*\* | .725\*\* | .614\* | 1 | | | | | | |
| **Cl** | .549 | **.851**\*\* | **.913**\*\* | **.880**\*\* | .853\*\* | **.768**\*\* | **.830**\*\* | 1 | | | | | |
| **SO$_4$** | .661 | .490 | .608\* | .399 | .484 | .661\*\* | .421 | .545\* | 1 | | | | |
| **CO$_3$** | .269 | -.005 | .186 | .135 | .382 | .123 | .112 | .173 | .212 | 1 | | | |
| **HCO$_3$** | -.470 | **.774**\*\* | **.783**\*\* | .727\*\* | .470 | .732\*\* | .677\*\* | .582\* | .182 | -.132 | 1 | | |
| **NO$_3$** | -.015 | .694\*\* | **.797**\*\* | .586\* | .732\*\* | .739\*\* | .710\*\* | .714\*\* | .633\* | .450 | .457 | 1 | |
| **F** | -.319 | -.171 | -.329 | -.180 | -.265 | -.310 | -.179 | -.355 | -.646\*\* | -.523\* | .110 | -.397 | 1 |

\* Correlation is significant at the 0.05 level (2-tailed) and

\*\*Correlation is significant at the 0.01 level (2-tailed).

The significance of "bold" emphasis that values are showing strong correlation between the corresponding parameters.

**Table 3. Inter-correlations matrix of different water quality parameters for POM season.**

| Parameter | pH | EC | TDS | Na | K | Ca | Mg | Cl | SO$_4$ | CO$_3$ | HCO$_3$ | NO$_3$ | F |
|---|---|---|---|---|---|---|---|---|---|---|---|---|---|
| pH | 1 | | | | | | | | | | | | |
| EC | -.125 | 1 | | | | | | | | | | | |
| TDS | .071 | .404 | 1 | | | | | | | | | | |
| Na | -.018 | -.052 | **.785***  | 1 | | | | | | | | | |
| K | .181 | -.332 | -.225 | -.004 | 1 | | | | | | | | |
| Ca | .003 | .266 | **.821*** | .699* | .363 | 1 | | | | | | | |
| Mg | -.104 | .548* | **.776*** | **.740*** | -.509 | .075 | 1 | | | | | | |
| Cl | .135 | **.809**** | **.717*** | **.803**** | -.165 | .693* | .544 | 1 | | | | | |
| SO$_4$ | .206 | .791** | -.479 | .098 | .694* | -.165 | -.694** | -.584* | 1 | | | | |
| CO$_3$ | -.069 | -.274 | .609 | .037 | .156 | .111 | -.061 | -.219 | .229 | 1 | | | |
| HCO$_3$ | -.096 | -.389 | -.052 | .620 | .004 | .094 | -.131 | -.286 | .355 | .215 | 1 | | |
| NO$_3$ | -.144 | .208 | .568 | .150 | .017 | .387 | .150 | .215 | -.233 | -.575* | -.181 | 1 | |
| F | -.004 | -.340 | -.253 | .059 | -.004 | -.755** | .027 | -.613* | .172 | .247 | -.196 | -.474 | 1 |

* Correlation is significant at the 0.05 level (2-tailed) and

**Correlation is significant at the 0.01 level (2-tailed).

The significance of "bold" emphasis that values are showing strong correlation between the corresponding parameters.

**Table 4. K-M-O and Barlett's test result.**

| K-M-O and B test | | PRM season | POM season |
|---|---|---|---|
| K-M-O adequacy | | 0.702 | 0.698 |
| Barlett's test Spherecity | Chi square | 855.23 | 789.95 |
| | df | 59 | 59 |
| | Sig | 0 | 0 |

factor analysis (PCs). The ig value of a relevant variable defines its peak value. Those eigenvalues with a magnitude >1 are the most essential factor. The components with ig that were <1 were disregarded because they lacked significance [71,95]. A scree plot shows (Fig 7a and 7b) that the ig values of the first four elements are all > 1. As of the 5th ig value, the slope gradually becomes flatter. Only the first four components have thus far been settled upon, with a PRM cumulative variance (%) of 91.30 and a POM variance (%) of 70.44.

The four PCA components, which were extracted based on eigenvalue greater than 1, explain 57.92%, 17.38%, 7.77%, and 8.23% of the total variance, respectively. Each component contains some strong positive, negative, and approximately zero loadings. The results of PCA for GWQ parameters based on the varimax rotation matrix are presented in Table 5. The first component explains about 57.92% of the total variance and has strong loading for EC, TDS, Na, Mg, and Cl, a moderate loading for K, HCO3, NO3, and Ca, and weak loading values for SO4. Hence, Component 1st is referred to as the "salinity" component of strong loadings between Na and Cl ions. Moderate loading between K and NO3 indicates the incorporation of chemical fertilizers and animal waste into agricultural activities in the study area [72,96]. Since no sanitation network was present during the sampling time, there were instances in which untreated domestic sewage was directly discharged into the aquifers. The component plot in a rotated space is depicted in Fig 8.

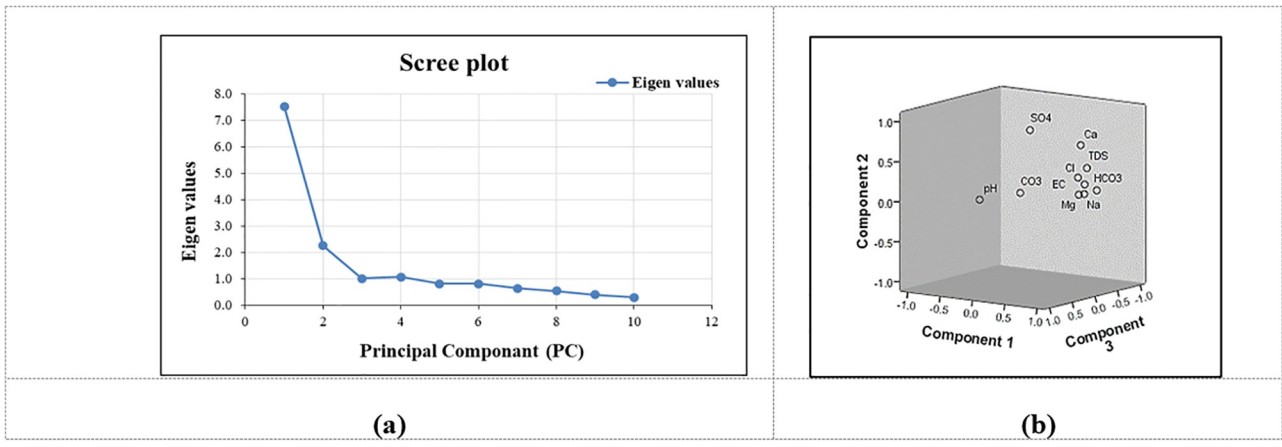

**Fig 7. Scree and component plot in rotated space for PC's in PRM.**

The 2$^{nd}$ component explained nearly 17.38% of the variance and was attributed to strong positive loading for sulfate (SO4), which corresponded to anthropogenic activities in the field, such as crop fertilization and other land-use activities, and weak loading for NO3. The 4th component explains the 8.23% of the variance, along with strong positive loadings for TDS and CO3. The PCA highlighted the order of importance of parameters viz., TDS> pH> EC> Na> Ca> Cl > Mg > CO3 > SO4> HCO3 in PRM season.

The PCA was applied to a WQ variable's correlation matrix consisting of 13 physio-chemical parameters. Its purpose is to determine the individual loadings of each of the 13 variables that affect WQ characteristics. Eigenvalues (ig) are frequently employed to derive components factor analysis (PCs). The ig value of a relevant variable defines its peak value. Those

**Table 5. Rotated component matrix obtained by PC analysis for PRM season.**

| Parameters | Principal Component | | | |
|---|---|---|---|---|
| | 1 | 2 | 3 | 4 |
| pH | -.003 | .215 | -.953 | .096 |
| EC | .942 | .176 | .034 | -.097 |
| TDS | .912 | .338 | .184 | .100 |
| Na | .925 | .079 | -.018 | .030 |
| K | .772 | .204 | .057 | .401 |
| Ca | .697 | .500 | .460 | .056 |
| Mg | .919 | .080 | -.080 | .071 |
| Cl | .896 | .278 | -.017 | .129 |
| SO$_4$ | .377 | .880 | -.151 | .003 |
| CO$_3$ | .047 | .179 | -.123 | .947 |
| HCO$_3$ | .779 | -.045 | .496 | -.173 |
| NO$_3$ | .665 | .416 | .151 | .412 |
| F | -.046 | -.795 | .170 | -.376 |
| Eigen value | 7.53 | 2.26 | 1.01 | 1.07 |
| Variance (%) | 57.92 | 17.38 | 07.77 | 08.23 |
| Cumulative Variance (%) | 57.92 | 75.30 | 83.07 | 91.30 |

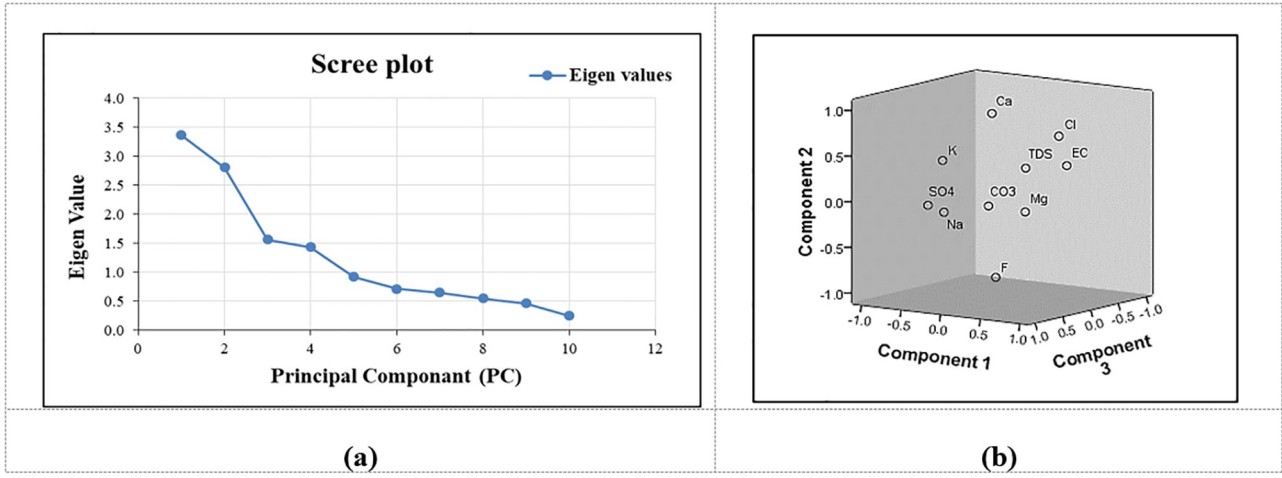

**Fig 8. Scree and component plot in rotated space for PC's in POM.**

eigenvalues with a magnitude >1 are the most essential factor. The components with ig that were <1 were disregarded because they lacked significance [64]. A scree plot shows (Fig 7) that the ig values of the first four elements are all > 1. As of the 5th ig value, the slope gradually becomes flatter. Only the first four components have thus far been settled upon, with a PRM cumulative variance (%) of 91.30 and a POM variance (%) of 70.44.

The four PCA components, which were extracted based on eigenvalue greater than 1, explain 57.92%, 17.38%, 7.77%, and 8.23% of the total variance, respectively. Each component contains some strong positive, negative, and approximately zero loadings. The results of PCA for GWQ parameters based on the varimax rotation matrix are presented in Table 5. The first component explains about 57.92% of the total variance and has strong loading for EC, TDS, Na, Mg, and Cl, a moderate loading for K, HCO3, NO3, and Ca, and weak loading values for SO4. Hence, Component 1st is referred to as the "salinity" component of strong loadings between Na and Cl ions. Moderate loading between K and NO3 indicates the incorporation of chemical fertilizers and animal waste into agricultural activities [65] in the study area. Since no sanitation network was present during the sampling time, there were instances in which untreated domestic sewage was directly discharged into the aquifers. The component plot in a rotated space is depicted in Fig 8.

The 2nd component explained nearly 17.38% of the variance. It was attributed to strong positive loading for sulfate (SO4), which corresponded to anthropogenic activities in the field, such as crop fertilization and other land-use activities, and weak loading for NO3. The 4th component explains the 8.23% of the variance, along with strong positive loadings for TDS and CO3. The PCA highlighted the order of importance of parameters viz., TDS> pH> EC> Na> Ca> Cl > Mg > CO3 > SO4> HCO3 in PRM season.

In POM season, the analysis indicated that 25.85% of the entire variance was explained by PC1 in the datasets with strong loading values for Mg and $NO_3$, moderate loading values for EC and TDS, weak loading values for Cl, which were mostly distributed between upper and central regions of JRB. This component reveals that rock–water interaction with ions exchange was responsible for the geogenic hydro-geochemical evolution of groundwater [6]. Das et al. [3] has also reported that the origin of salinity (presence of Na-Cl) in croplands was primarily due to the presence of chemical fertilizers, animal waste and industrial pollutants. Moreover,

this factor also has moderate loading between K and $NO_3$ indicating that the cultivating activities and industrial disposal near the water source in the study area.

The PC2 component explicates nearly 21.55% of the entire variance (Table 6). The Ca content exhibited strong positive loading values, which indicates the influence of 'hardness' associated with presence of carbonates in groundwater, rendering it unfit for drinking and irrigation purposes and it is also responsible for weak loading of $NO_3$. The 3rd component had positive coefficients for Na and $HCO_3$ which explained for only 12.02% of the entire variation. This component is usually associated with seepage or untreated sewage water to the groundwater. The 4th PCA component explains the 11.02% of the entire variation along with strong positive loading for $CO_3$.

**3.2.3 Hierarchical Cluster Analysis (HCA).** In the HCA, water quality parameters with higher degrees of similarity were assigned to the first cluster. Based on Fig 9, two main clusters were found among the water quality parameters. The first cluster consisted of four parameters viz., EC, TDS, Na, and Ca, which might be influenced by the mix sources, over-pumping of groundwater, dissolution of alkaline rocks, and leaching of fertilizers from the soil horizon to the aquifer. The second cluster consists of three parameters, i.e., K, Cl, and Mg. The second cluster is characterized by anthropogenic sources such as agricultural operations, sewage waste, and drainage water infiltration from bleaching industries. It also became apparent that pH, F, and SO4 could not be clustered properly with other clusters.

Similarly, two main clusters were found among the water quality parameters for the post-monsoon season. The first cluster consisted of three parameters, viz., EC, TDS, and Cl, whereas the second cluster consisted of nine parameters, among which Na and HCO3, K, and SO4 were found to be closely related (Fig 10). Based on the observations made through the dendrogram and correlation matrix, the TDS and EC parameters were strongly correlated with respect to Ca, Na, and Cl. The pH parameter, however, did not demonstrate any significant association with other water quality parameters in the post-monsoon season [47]. These HCA results were found to agree with the correlation matrix and PCA.

**Table 6. Rotated component matrix obtained by PC analysis for POM season.**

| Parameters | Principal Component | | | |
|---|---|---|---|---|
| | 1 | 2 | 3 | 4 |
| pH | -.182 | .116 | -.245 | .176 |
| EC | .761 | .360 | -.307 | -.166 |
| TDS | .556 | .558 | .205 | .342 |
| Na | .149 | .020 | .852 | -.031 |
| K | -.717 | .291 | -.065 | .104 |
| Ca | -.035 | .953 | .065 | .056 |
| Mg | .857 | .055 | .397 | .017 |
| Cl | .476 | .641 | -.449 | -.095 |
| $SO_4$ | -.898 | -.172 | .185 | .139 |
| $CO_3$ | -.100 | .031 | .149 | .925 |
| $HCO_3$ | -.338 | .101 | .619 | .164 |
| $NO_3$ | .100 | .439 | .181 | -.734 |
| pH | -.182 | .116 | -.245 | .176 |
| Eigen value | 3.36 | 2.80 | 1.56 | 1.43 |
| Variance (%) | 25.85 | 21.55 | 12.02 | 11.02 |
| Cumulative Variance (%) | 25.85 | 47.40 | 58.42 | 70.44 |

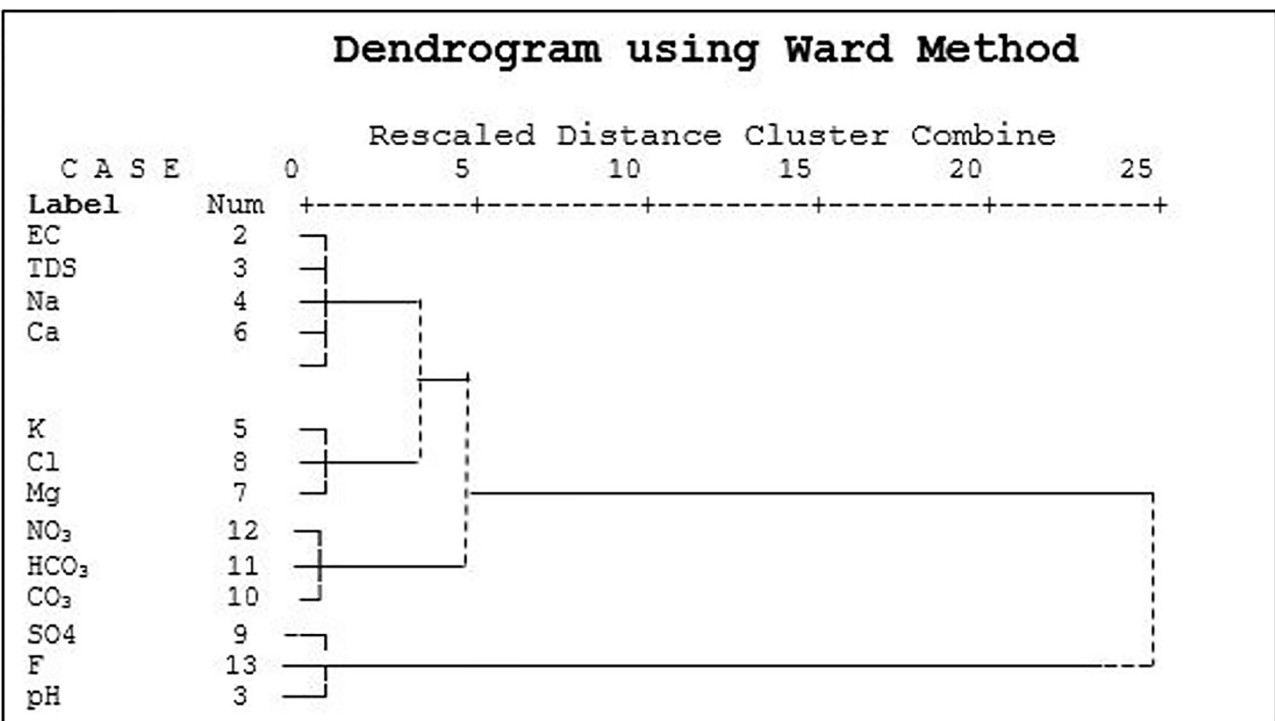

**Fig 9. Dendrogram presenting clustering of GWQ parameters for pre-monsoon.**

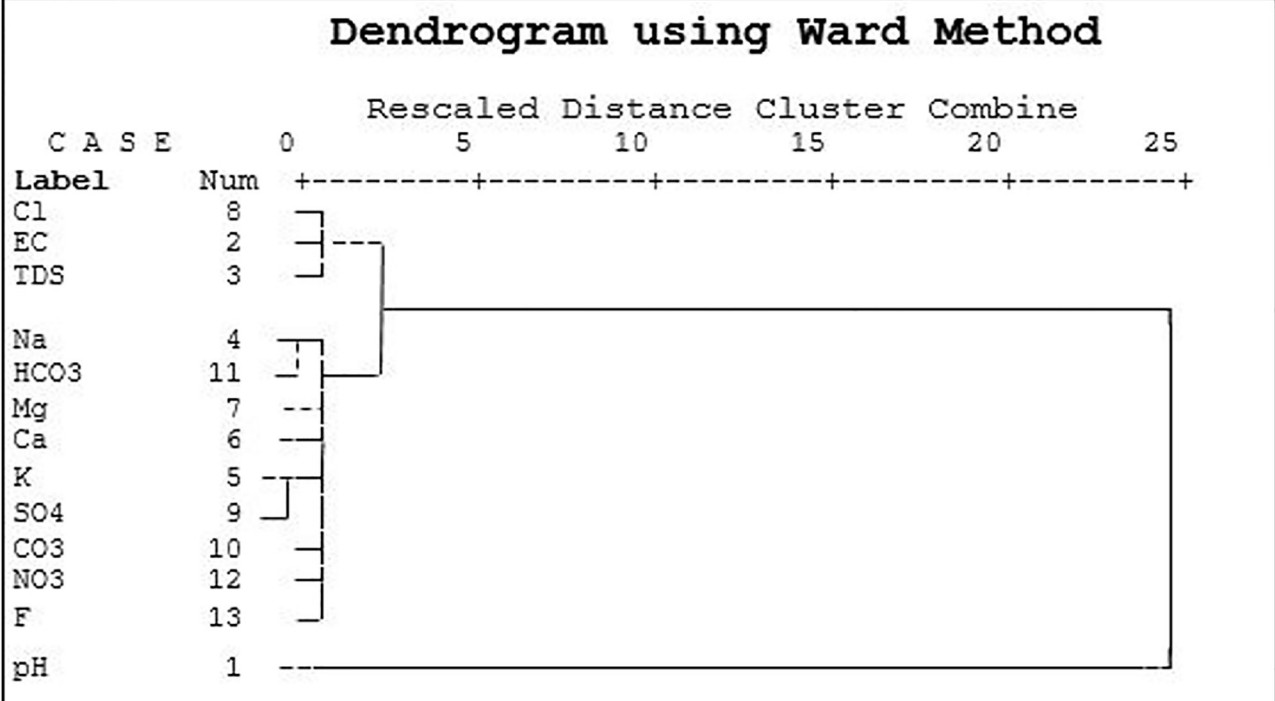

**Fig 10. Dendrogram presenting clustering of GWQ parameters for post-monsoon.**

## 4. Conclusions

This study attempts to characterize and interpret the GWQ under the integrated approach of GIS and multivariate statistical approach (MSA) for the Jakham River Basin (JRB). As per the GIS-based WQI findings, 63.42 percent of the groundwater samples during the PRM) season and 42.02 percent during the POM were classed as 'good' and could be deemed fit for human consumption. Nevertheless, imparting a spatial sense to the WQ of local aquifers through mapping of WQI highlighted the higher scale of WQI values in northern and central regions of the JRB, whereas poor WQ was observed in the lower areas of the basin. The findings of the PCA showed that anthropogenic and geogenic causes cause changes in the physio-chemical properties of groundwater strata. In addition, the correlation matrix (CM) outcomes show a strong stake in the conclusions generated by PCA and CA analyses. In PCA analysis, four components account for 91.30 percent and 70.44 percent of the total variance for PRM and POM season; these four components have executed as significant quality controlling factors. The evaluation of the GWQ through statistical tools and graphical representation assists in detecting the most vulnerable zones, which deteriorate the GWQ and negatively impact consumers' health. The spatial maps of the WQI index will help to detect the real-time water quality for drinking and irrigation purposes. Also, the water health card can be generated based on the study outcomes. Therefore, this study's outcomes may help water resource planners and policymakers prioritize and safeguard the groundwater supply and develop technology that maintains groundwater quality.

## Acknowledgments

The authors extend their appreciation to the Deputyship for Research and Innovation, Ministry of Education in Saudi Arabia for funding this research work through the project no. (IFK-SUOR3- 622).

## Author Contributions

**Conceptualization:** Vinay Kumar Gautam, Chaitanya B. Pande, Fahad Alshehri, Zaher Mundher Yaseen.

**Data curation:** Mahesh Kothari, Pradeep Kumar Singh.

**Formal analysis:** Mahesh Kothari, Baqer Al-Ramadan, Chaitanya B. Pande, Fahad Alshehri.

**Funding acquisition:** Zaher Mundher Yaseen.

**Investigation:** Vinay Kumar Gautam, Mahesh Kothari, Baqer Al-Ramadan, Pradeep Kumar Singh, Harsh Upadhyay, Chaitanya B. Pande, Fahad Alshehri.

**Methodology:** Vinay Kumar Gautam, Harsh Upadhyay, Chaitanya B. Pande.

**Project administration:** Zaher Mundher Yaseen.

**Supervision:** Zaher Mundher Yaseen.

**Validation:** Vinay Kumar Gautam, Pradeep Kumar Singh, Chaitanya B. Pande, Zaher Mundher Yaseen.

**Visualization:** Vinay Kumar Gautam, Baqer Al-Ramadan, Harsh Upadhyay, Fahad Alshehri.

**Writing – original draft:** Vinay Kumar Gautam, Mahesh Kothari, Chaitanya B. Pande, Fahad Alshehri.

**Writing – review & editing:** Vinay Kumar Gautam, Mahesh Kothari, Baqer Al-Ramadan, Pradeep Kumar Singh, Harsh Upadhyay.

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
