## [Decision Letter · Decision Letter 0]

17 Aug 2023

PONE-D-23-19872Groundwater quality characterization with the integration of water quality index and multivariate statistical techniquesPLOS ONE

Dear Dr. Pande,

Thank you for submitting your manuscript to PLOS ONE. After careful consideration, we feel that it has merit but does not fully meet PLOS ONE’s publication criteria as it currently stands. Therefore, we invite you to submit a revised version of the manuscript that addresses the points raised during the review process.

We look forward to receiving your revised manuscript.

Kind regards,

Sani Isah Abba, PhD

Academic Editor

PLOS ONE

“The authors extend their appreciation to Abdullah Alrushaid Chair for Earth Science Remote Sensing Research for funding.”

“No”

5. Please include a caption for figure 6.

6. We note that Figures 1, 2 and 5 in your submission contain [map/satellite] images which may be copyrighted. All PLOS content is published under the Creative Commons Attribution License (CC BY 4.0), which means that the manuscript, images, and Supporting Information files will be freely available online, and any third party is permitted to access, download, copy, distribute, and use these materials in any way, even commercially, with proper attribution. For these reasons, we cannot publish previously copyrighted maps or satellite images created using proprietary data, such as Google software (Google Maps, Street View, and Earth). For more information, see our copyright guidelines: http://journals.plos.org/plosone/s/licenses-and-copyright.

1. You may seek permission from the original copyright holder of Figures 1, 2 and 5 to publish the content specifically under the CC BY 4.0 license. 

Reviewers' comments:

Reviewer's Responses to Questions

**Comments to the Author**

1. Is the manuscript technically sound, and do the data support the conclusions?

Reviewer #1: Yes

Reviewer #2: Yes

2. Has the statistical analysis been performed appropriately and rigorously? 

Reviewer #1: Yes

Reviewer #2: Yes

3. Have the authors made all data underlying the findings in their manuscript fully available?

Reviewer #1: Yes

Reviewer #2: Yes

4. Is the manuscript presented in an intelligible fashion and written in standard English?

Reviewer #1: Yes

Reviewer #2: Yes

5. Review Comments to the Author

Reviewer #1: The authors have presented a “Groundwater quality characterization with the integration of water quality index and multivariate statistical techniques”. Few correction and suggestion before it can be accepted.

1. How many year data was used for the study.

2. Please indicate the novelty of the work.

3. Check Eq 2 for typo error.

4. Figure 6 is not stated in the text.

5. Figure 6 is not readable.

6. Do these many figures are really needed or can you can add in supplementary data? Also remove the water mark from the figure 6.

Reviewer #2: The work by Gautam et al. deals with the characterization and interpretation of the groundwater quality (GWQ)

using GIS environment and multivariate statistical approach (MSA) for Jakham River Basin (JRB) in the Southern Rajasthan. The topic is of great interest with the widespread application. Therefore, it is suggested for publication in the PLOS one journal, after a minor/moderate revision, which should include the following suggestions:

1- It would be beneficial to elaborate on the distinctive contributions of this research, especially in light of similar studies conducted previously. How does your work stand apart in terms of novelty?

2-How can the insights from this research be operationalized into tangible measures or technological solutions to enhance and safeguard groundwater quality?

3-The decision to employ HCA over other advanced clustering techniques needs further justification. While its common use is acknowledged, the rationale for its selection in this specific context should be clear. Could you elaborate on this choice?

4-In determining the WQI using GIS, what criteria or parameters did you adopt to classify groundwater samples as 'good'?

5-you should explain why the integrated approach of GIS and multivariate statistical approach (MSA) enhance the characterization of groundwater quality (GWQ) compared to using either method alone?

6-The concluding section appears to have an abrupt ending. Kindly revise it.

7-In conclusion section you should revise it based on the study's findings, what specific recommendations can be made for water resource planners and policymakers in the JRB?

8-I noticed a potential issue in Equation 2; it appears there might be a missing parenthesis on the upper side. Could you please verify this?

9- Figure 6 doesn't have caption. you must provide a detailed description for it.

Please double check the caption of figure 7 and 8 as well, you must mention the caption of each subplot as (a), and (b) separately.

6. PLOS authors have the option to publish the peer review history of their article (what does this mean?). If published, this will include your full peer review and any attached files.

Reviewer #1: No

Reviewer #2: No

---

## [Author Response · Author response to Decision Letter 0]

12 Oct 2023

Author response to reviewer’s comments: 

Reply: Yes, we have included as per suggestion.

Reply: Thanks for suggestion, no any code created by authors. All information shared in the article.

“The authors extend their appreciation to Abdullah Alrushaid Chair for Earth Science Remote Sensing Research for funding.”

“No”

Reply: Yes, funding statement is removed as per suggestion. Added in the Funded section.

Reply: we have revised the data availability statement.

5. Please include a caption for figure 6.

Reply: Thanks for suggestions, we have included a caption of Fig.6.

6. We note that Figures 1, 2 and 5 in your submission contain [map/satellite] images which may be copyrighted. All PLOS content is published under the Creative Commons Attribution License (CC BY 4.0), which means that the manuscript, images, and Supporting Information files will be freely available online, and any third party is permitted to access, download, copy, distribute, and use these materials in any way, even commercially, with proper attribution. For these reasons, we cannot publish previously copyrighted maps or satellite images created using proprietary data, such as Google software (Google Maps, Street View, and Earth). For more information, see our copyright guidelines: http://journals.plos.org/plosone/s/licenses-and-copyright.

1. You may seek permission from the original copyright holder of Figures 1, 2 and 5 to publish the content specifically under the CC BY 4.0 license. 

Reply: All the figures are prepared by the author itself. There is no copying from other sources. This is the PhD research work of the other, therefore all the map copy right reserved by the author. The author has already submitted this, on thesis.

Comments to the Author

1. Is the manuscript technically sound, and do the data support the conclusions?

Reviewer #1: Yes

Reviewer #2: Yes

Reply: Thanks for appreciated the work.

2. Has the statistical analysis been performed appropriately and rigorously?

Reviewer #1: Yes

Reviewer #2: Yes

Reply: Thanks for appreciated the work.

3. Have the authors made all data underlying the findings in their manuscript fully available?

Reviewer #1: Yes

Reviewer #2: Yes

Reply: Thanks for appreciated the work.

4. Is the manuscript presented in an intelligible fashion and written in standard English?

Reviewer #1: Yes

Reviewer #2: Yes

Reply: Thanks for appreciated the work.

5. Review Comments to the Author

Reviewer #1: 

The authors have presented a “Groundwater quality characterization with the integration of water quality index and multivariate statistical techniques”. Few correction and suggestion before it can be accepted.

Reply: Thanks for suggestion and accepted the article for publication. As per you suggestion corrections were made in the revised article.

1. How many year data was used for the study.

Reply: Total 13 years of data have been used.

2. Please indicate the novelty of the work.

Reply: The study has been conducted in hard rock and tribal are of southern Rajasthan, which have more water quality problem as well as the water availability. Only the govt. agencies are working in this area, which is not sufficiently monitored. Therefore water quality was analyzed using previous data with multivariate statistical analysis and Geospatial technique, which is novel approach for these type of semi-arid hard rock region.

3. Check Eq 2 for typo error.

Reply: Corrected and improved

4. Figure 6 is not stated in the text.

Reply: added as per suggestion

5. Figure 6 is not readable.

Reply: These fig. are direct output of the analysis software. These are good quality graphs. Due to lower size, these have less visibility, but the size can be expanded as per requirement.

6. Do these many figures are really needed or can you can add in supplementary data? Also remove the water mark from the figure 6.

Reply: Yes, these graphs represents the mean, max and min value of the water quality parameters of the study area. Therefore it is necessary to present in main text of the manuscript.

Reviewer #2: 

The work by Gautam et al. deals with the characterization and interpretation of the groundwater quality (GWQ) using GIS environment and multivariate statistical approach (MSA) for Jakham River Basin (JRB) in the Southern Rajasthan. The topic is of great interest with the widespread application. Therefore, it is suggested for publication in the PLOS one journal, after a minor/moderate revision, which should include the following suggestions:

Reply: Thanks for suggestion and accepted the article for publication. As per you suggestion corrections were made in the revised article.

1- It would be beneficial to elaborate on the distinctive contributions of this research, especially in light of similar studies conducted previously. How does your work stand apart in terms of novelty?

Reply: The study has been conducted in hard rock and tribal are of southern Rajasthan, which have more water quality problem as well as the water availability. Only the govt. agencies are working in this area, which is not sufficiently monitored. Therefore water quality was analyzed using previous data with multivariate statistical analysis and Geospatial technique, which is novel approach for these type of semi-arid hard rock region.

2-How can the insights from this research be operationalized into tangible measures or technological solutions to enhance and safeguard groundwater quality?

Reply: In the further recommendation of this research Decision support system are proposed for live detection and prediction of water quality in village level. Which will help to safeguard the potable water as well irrigation water.

3-The decision to employ HCA over other advanced clustering techniques needs further justification. While its common use is acknowledged, the rationale for its selection in this specific context should be clear. Could you elaborate on this choice?

Reply: As a data-driven method, HCA operates without a priori assumptions about aquifer geology, confinement, style and rate of water–rock interaction, or any other factors that might control the categorization. Another strength of HCA is that it can be based on any number of variables, and these variables can be of any type (chemical, physical, distributed or non-distributed). Therefore, in this study HCA was used as a cluster analysis tool.

4-In determining the WQI using GIS, what criteria or parameters did you adopt to classify groundwater samples as 'good'?

Reply: Total 13 parameters were identified for water quality indexing. Based on the analysis WQI value were determined for each observed well in the study area. The criteria based on the WQI single value given here: I (0-25)- Excellent, II (26-50)- Good, III (51-75)-Poor, IV (76-100)- Very Poor, V (> 100)- Unfit for consumption

5-you should explain why the integrated approach of GIS and multivariate statistical approach (MSA) enhance the characterization of groundwater quality (GWQ) compared to using either method alone?

Reply: multivariate statistical approach (MSA) helps to analyses the time series data of water quality, continuously these data can be applied as an input in the GIS tools for spatial mapping of groundwater quality. Therefore, this integrated technique is more precise than alone.

6-The concluding section appears to have an abrupt ending. Kindly revise it.

Reply: As per suggestion conclusion is revised.

7-In conclusion section you should revise it based on the study's findings, what specific recommendations can be made for water resource planners and policymakers in the JRB?

Reply: Based on the study, a consistent investigation of groundwater quality characteristics will reduce the prospect for further deterioration of soil and human health. However, the outcome of the present result may help to comprehend the excellence of the groundwater resources to improve appropriate management plan. Spatial map of WQI index will help to detect the live water quality for drinking as well irrigation purpose. Also the water health card can be generated based the predicted and present data. Added and improved as per suggestion.

8-I noticed a potential issue in Equation 2; it appears there might be a missing parenthesis on the upper side. Could you please verify this?

Reply: As per suggestion, equation have been improved and corrected.

9- Figure 6 doesn't have caption. you must provide a detailed description for it.

Please double check the caption of figure 7 and 8 as well, you must mention the caption of each subplot as (a), and (b) separately.

Reply: The caption provided in the fig.6. All the fig represents the box plot of water quality parameters, where (a) section displays pre monsoon data and (b) displays post monsoon season data. The details regarding graph already mentioned in the main text.

6. PLOS authors have the option to publish the peer review history of their article (what does this mean?). If published, this will include your full peer review and any attached files.

Do you want your identity to be public for this peer review? For information about this choice, including consent withdrawal, please see our Privacy Policy.

Reviewer #1: No

Reviewer #2: No

Reply: Thanks for suggestions.

---

## [Decision Letter · Decision Letter 1]

3 Nov 2023

Groundwater quality characterization using an integrated water quality index and multivariate statistical techniques

PONE-D-23-19872R1

Dear Dr. Pande,

We’re pleased to inform you that your manuscript has been judged scientifically suitable for publication and will be formally accepted for publication once it meets all outstanding technical requirements.

Kind regards,

Sani Isah Abba, PhD

Academic Editor

PLOS ONE

Additional Editor Comments (optional):

Reviewers' comments:

Reviewer's Responses to Questions

**Comments to the Author**

1. If the authors have adequately addressed your comments raised in a previous round of review and you feel that this manuscript is now acceptable for publication, you may indicate that here to bypass the “Comments to the Author” section, enter your conflict of interest statement in the “Confidential to Editor” section, and submit your "Accept" recommendation.

Reviewer #1: All comments have been addressed

2. Is the manuscript technically sound, and do the data support the conclusions?

Reviewer #1: Yes

3. Has the statistical analysis been performed appropriately and rigorously? 

Reviewer #1: Yes

4. Have the authors made all data underlying the findings in their manuscript fully available?

Reviewer #1: Yes

5. Is the manuscript presented in an intelligible fashion and written in standard English?

Reviewer #1: Yes

6. Review Comments to the Author

Reviewer #1: I have carefully reviewed all the comments, and I am pleased to recommend this for acceptance. The feedback and suggestions provided have been thoroughly addressed, resulting in a significantly improved.

7. PLOS authors have the option to publish the peer review history of their article (what does this mean?). If published, this will include your full peer review and any attached files.

Reviewer #1: No

---

## [Editor Report · Acceptance letter]

18 Dec 2023

PONE-D-23-19872R1 

PLOS ONE

Dear Dr. Pande, 

I'm pleased to inform you that your manuscript has been deemed suitable for publication in PLOS ONE. Congratulations! Your manuscript is now being handed over to our production team.

Kind regards, 

on behalf of

Dr. Sani Isah Abba 

%CORR_ED_EDITOR_ROLE%

PLOS ONE